# Generalizable Origin Identification for Text-Guided Image-to-Image Diffusion Models

*Change this image to President Joe Biden being assassinated.*

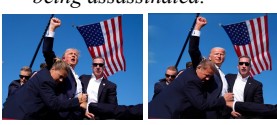

*Remove the watermark. An overhead view of a couple walking hand in hand along a narrow sandbar.*

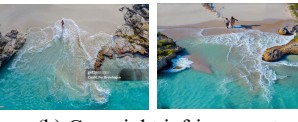

*Translate the image of a modern urban scene with a wrecked car into a medieval setting.*

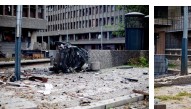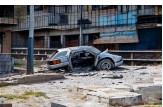

(a) Misinformation  (b) Copyright infringement  (c) Evading content tracing

*Figure 1:* The illustration for **misusing** text-guided image-to-image diffusion models in several scenarios: *misinformation*, *copyright infringement*, and *evading content tracing*. Specifically: **(a)** An altered image originally showing Donald Trump post-assassination is edited to depict Joe Biden instead; **(b)** The removal of a watermark from a copyrighted beach image, followed by modifications, could assist in escaping copyright checks; **(c)** An image of a Norwegian government building after an explosion is altered to bypass restrictions, which limit the spread of disturbing images.

## Abstract

Text-guided image-to-image diffusion models excel in translating images based on textual prompts, allowing for precise and creative visual modifications. However, such a powerful technique can be misused for *spreading misinformation*, *infringing on copyrights*, and *evading content tracing*. This motivates us to introduce the task of origin **ID**entification for text-guided **I**mage-to-image **D**iffusion models (**ID$^2$**), aiming to retrieve the original image of a given translated query. A straightforward solution to ID$^2$ involves training a specialized deep embedding model to extract and compare features from both query and reference images. However, due to *visual discrepancy* across generations produced by different diffusion models, this similarity-based approach fails when training on images from one model and testing on those from another, limiting its effectiveness in real-world applications. To solve this challenge of the proposed ID$^2$ task, we contribute the first dataset and a theoretically guaranteed method, both emphasizing generalizability. The curated dataset, **OriPID**, contains abundant **Ori**gins and guided **P**rompts, which can be used to train and test potential **ID**entification models across various diffusion models. In the method section, we first prove the *existence* of a linear transformation that minimizes the distance between the pre-trained Variational Autoencoder (VAE) embeddings of generated samples and their origins. Subsequently, it is demonstrated that such a simple linear transformation can be *generalized* across different diffusion models. Experimental results show that the proposed method achieves satisfying generalization performance, significantly surpassing similarity-based methods (+31.6% mAP), even those with domain generalization designs.

## 1 Introduction

Text-guided image-to-image diffusion models are notable for their ability to transform images based on textual descriptions, allowing for detailed and highly customizable modification. While they are increasingly used in creative industries for tasks such as digital art re-creation, customizing visual content, and personalized virtual try-ons, there are growing security concerns associated with their **misuse**. As illustrated in Fig. 1, for instance, they could be misused for *misinformation*, *copyright infringement*, and *evading content tracing*. To help combat these misuses, this paper introduces the task of origin **ID**entification for text-guided **I**mage-to-image **D**iffusion models (**ID$^2$**), which aims to

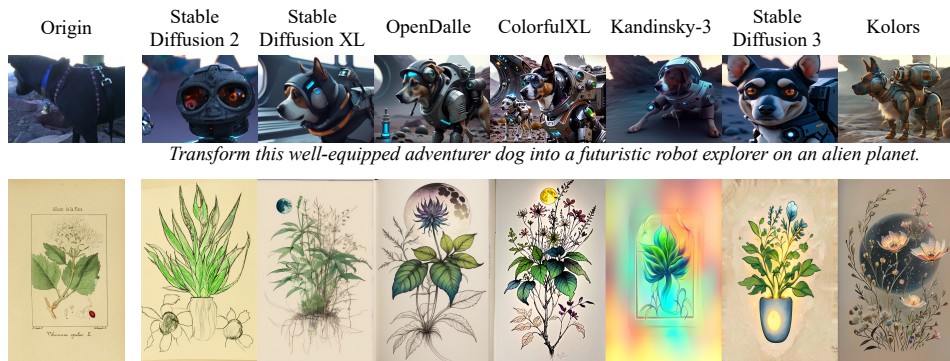

*Transform this well-equipped adventurer dog into a futuristic robot explorer on an alien planet.*

*Transform the delicate botanical sketch into a vibrant, fantastical plant glowing under a moonlit night.*

*Figure 2:* The demonstration for *visual discrepancy* between generated images by different diffusion models. The images generated by various models exhibit distinctive visual features such as realistic textures, complex architectures, life-like details, vibrant colors, abstract expression, magical ambiance, and photorealistic elements.

identify the original image of a generated query from a large-scale reference set. When the origin is identified, subsequent compensations include deploying factual corrections for misinformation, enforcing copyright compliance, and keeping the tracing of target content.

A straightforward solution for the proposed $ID^2$ task is to employ a similarity-based retrieval approach. Specifically, this approach **(1)** fine-tunes a pre-trained network by minimizing the distances between generated images and their origins, and **(2)** uses the trained network to extract and compare feature vectors from the queries and references. However, this approach is **impractical** in real-world scenarios. This is because: for most current popular diffusion models, such as Stable Diffusion 2 (Rombach et al., 2022), Stable Diffusion XL (Podell et al., 2024), OpenDalle (Izquierdo, 2023), ColorfulXL (Recoilme, 2023), Kandinsky-3 (Arkhipin et al., 2023), Stable Diffusion 3 (Esser et al., 2024), and Kolors (KolorsTeam, 2024), in a training-free manner, text-guided image-to-image translation can be easily achieved by using an input image with added noise as the starting point (instead of starting from randomly distributed noise). Further, as shown in Fig. 2, there exists a *visual discrepancy* across images generated by different diffusion models, *i.e.*, different diffusion models exhibit distinct visual features. An experimental evidence for such discrepancy is that we can train a lightweight classification model, such as Swin-S (Liu et al., 2021), to achieve a top-1 accuracy of **95.9%** when classifying images generated by these seven diffusion models. The visual discrepancy presents an inherent challenge of our $ID^2$, *i.e.*, *the approach mentioned above fails when trained on images generated by one diffusion model and tested on queries from another*. For instance, when trained on images generated by Stable Diffusion 2, this approach achieves a **87.1%** mAP on queries from Stable Diffusion 2, while only achieving a **30.5%** mAP on queries from ColorfulXL.

To address the generalization challenge in the proposed task, our efforts focus primarily on *constructing the first $ID^2$ dataset* and *proposing a method with theoretical guarantees*.

• **A new dataset emphasizing generalization.** To verify the generalizability, we construct the first $ID^2$ dataset, **OriPID**, which includes abundant **Ori**gins with guided **P**rompts for training and testing potential **ID**entification models. Specifically, the *training* set contains $100,000$ origins. For each origin, we use GPT-4o (OpenAI, 2024) to generate 20 different prompts, each of which implies a plausible translation direction. By inputting these origins and prompts into Stable Diffusion 2, we generate $2,000,000$ training images. For *testing*, we randomly select $5,000$ images as origins from a reference set containing $1,000,000$ images, and ask GPT-4o to generate a guided prompt for each origin. Subsequently, we generate $5,000$ queries using the origins, corresponding prompts, and each of the following models: Stable Diffusion 2, Stable Diffusion XL, OpenDalle, ColorfulXL, Kandinsky-3, Stable Diffusion 3, and Kolors. The design of using different diffusion models to generate training images and queries is particularly practical because, in the real world, where numerous diffusion models are publicly available, we cannot predict which ones might be misused.

• **A simple, generalizable, and theoretically guaranteed solution.** To solve the generalization problem, we first theoretically prove that, after specific linear transformations, the embeddings of an original image and its translation, encoded by the diffusion model's Variational Autoencoder

(VAE), will be sufficiently close. This suggests that we can use these linearly transformed query embeddings to match against the reference embeddings. Furthermore, we demonstrate that these kinds of feature vectors are generalizable across diffusion models. Specifically, by using a trained linear transformation and the encoder of VAE from one diffusion model, we can also effectively embed the generated images from another diffusion model, even if their VAEs have different parameters or architectures (see the Section 5.3 for more details). The effectiveness means the similar performance of origin identification for both diffusion models. Finally, we implement this theory (obtain the expected linear transformation) by gradient descending a metric learning loss and experimentally show the effectiveness and generalizability of the proposed solution.

In summary, this paper makes the following contributions:

1. This paper proposes a novel task, origin identification for text-guided image-to-image diffusion models ($ID^2$), which aims to identify the origin of a generated query. This task tries to alleviate an important and timely security concern, *i.e.*, the misuse of text-guided image-to-image diffusion models. To support this task, we build the first $ID^2$ dataset.

2. We highlight an inherent challenge of $ID^2$, *i.e.*, the existing visual discrepancy prevents similarity-based methods from generalizing to queries from unknown diffusion models. Therefore, we propose a simple but generalizable method by utilizing linear-transformed embeddings encoded by the VAE. Theoretically, we prove the existence and generalizability of the required linear transformation.

3. Extensive experimental results demonstrate (1) the challenge of the proposed $ID^2$ task: all pre-trained deep embedding models, fine-tuned similarity-based methods, and specialized domain generalization methods fail to achieve satisfying performance; and (2) the effectiveness of our proposed method: our UFC achieves 88.8%, 81.5%, 87.3%, 89.3%, 85.7%, 85.7%, and 90.3% mAP, respectively, for seven different diffusion models.

## 2 RELATED WORKS

**Diffusion Models.** Diffusion models have become a transformative class of generative models, utilizing iterative noise-based processes to excel in tasks such as image synthesis, inpainting, and text-to-image generation. By progressively denoising data, these models can reconstruct highly detailed images, offering flexibility and precision in creative applications. Recent advancements, including Stable Diffusion 2 (Rombach et al., 2022), Stable Diffusion XL (Podell et al., 2024), OpenDalle (Izquierdo, 2023), ColorfulXL (Recoilme, 2023), Kandinsky-3 (Arkhipin et al., 2023), Stable Diffusion 3 (Esser et al., 2024), and Kolors (KolorsTeam, 2024), have brought significant improvements in resolution, text-image alignment, and color dynamics. This paper considers using these popular diffusion models for text-guided image-to-image translation in a training-free manner, which is a common and cost-effective approach in the real world.

**Security Issues with AI-Generated Content.** Recently, generative models have gained significant attention due to their impressive capabilities. However, alongside their advancements, several security concerns have been identified. Prior research has explored various dimensions of these security issues. For instance, (Lin et al., 2024) focuses on detecting AI-generated multimedia to prevent its associated societal disruption. Additionally, (Ren et al., 2024) highlights the importance of verifying copyrighted material and addresses the legal challenges of safeguarding intellectual property rights for AI-generated works. Furthermore, (Fan et al., 2023) and (Chen et al., 2023) explore the ethical implications and technical challenges in ensuring the integrity and trustworthiness of AI-generated content. In contrast, while our work also aims to help address the security issues, we specifically focus on a novel perspective: identifying the origin of a given translated image.

**Image Copy Detection.** The task most similar to our $ID^2$ is Image Copy Detection (ICD) (Papakipos et al., 2022), which identifies whether a query *replicates the content* of any reference. Various works focus on different aspects: PE-ICD (Wang et al., 2024b) and AnyPattern (Wang et al., 2024a) build benchmarks and propose solutions emphasizing novel patterns in realistic scenarios; ASL (Wang et al., 2023a) addresses the hard negative challenge; Active Image Indexing (Fernandez et al., 2023) explores improving the robustness of ICD and retrieval by making imperceptible changes to images; and SSCD (Pizzi et al., 2022) leverages self-supervised contrastive learning to establish a strong

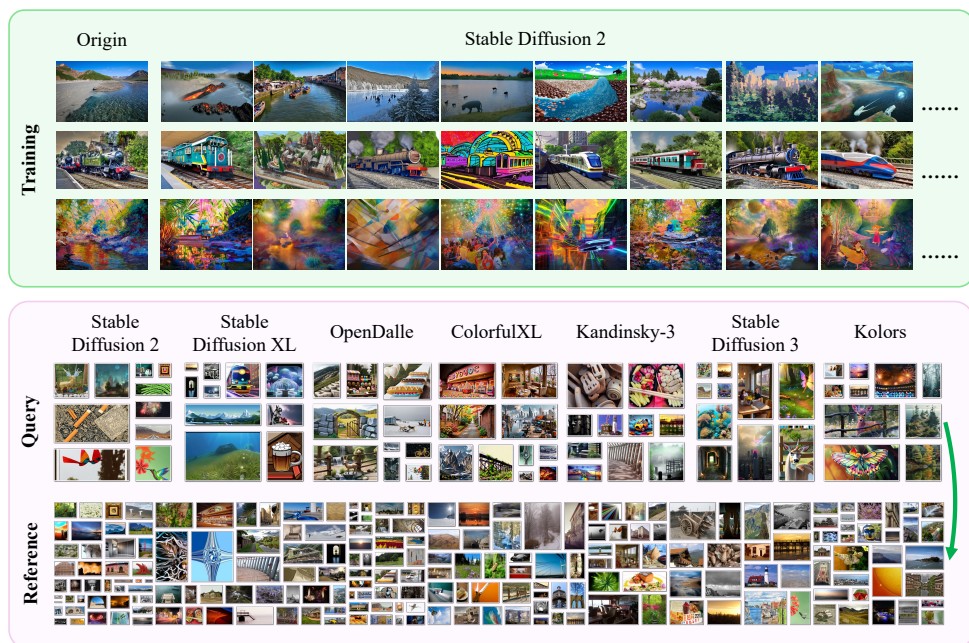

*Figure 3:* The images in our dataset, which is diverse and comprehensive. Specifically, it encompasses a variety of subjects commonly found in real-world scenarios where issues such as misinformation, copyright infringement, and content tracing evasion occur. For instance, our dataset includes images of nature, architecture, animals, planes, art, and indoor. Note that for simplicity, we omit the prompts here. Please refer to Appendix B for examples of prompts and generations.

baseline for ICD. Unlike ICD, which focuses on *manually-designed transformations*, our ID$^2$ aims to find the origin of a query translated by the *diffusion model with prompt-guidance*.

## 3 DATASET

To advance research in ID$^2$, this section introduces OriPID, the first dataset specifically designed for the proposed task. The source images in OriPID are derived from the DISC21 dataset (Papakipos et al., 2022), which is a subset of the real-world multimedia dataset YFCC100M (Thomee et al., 2016). As a result, OriPID is diverse and comprehensive, encompassing a wide range of subjects found in real-world scenarios where issues such as misinformation, copyright infringement, and content tracing evasion frequently occur. An illustration of the proposed dataset is shown in Fig. 3.

**Training Set.** The training set comprises (1) $100,000$ origins randomly selected from the $1,000,000$ original images in DISC21, (2) $2,000,000$ guided prompts (20 for each origin) generated by GPT-4o (for details on how these prompts were generated, see Appendix C), and (3) $2,000,000$ images generated by inputting the origins and prompts into Stable Diffusion 2 (Rombach et al., 2022).

**Test Set.** We design the test set with a focus on real-world/practical settings. On one hand, we use seven popular diffusion models, namely, Stable Diffusion 2 (Rombach et al., 2022), Stable Diffusion XL (Podell et al., 2024), OpenDalle (Izquierdo, 2023), ColorfulXL (Recoilme, 2023), Kandinsky-3 (Arkhipkin et al., 2023), Stable Diffusion 3 (Esser et al., 2024), and Kolors (KolorsTeam, 2024), to generate queries. This setting well simulates real-world scenarios where new diffusion models continuously appear, and we do not know which one is being misused. On the other hand, for each diffusion model, we generate $5,000$ queries to match $1,000,000$ references inherited from DISC21. This needle-in-a-haystack setting mimics the real world, where many distractors are not translated by any diffusion models.

**Scalability.** Currently, we only use Stable Diffusion 2 (Rombach et al., 2022) to generate training images. However, our OriPID can be easily scaled by incorporating more diffusion models for training, which may result in better generalizability. Furthermore, we only use $100,000$ origins and generate 20 prompts for each origin. Researchers are free to scale up our dataset by using the entire $1,000,000$ original images and generating more prompts with the script provided in Appendix C.

## 4 METHOD

To solve the proposed $ID^2$, we introduce a simple yet effective method, which is theoretically guaranteed and emphasizes generalizability. This section first presents two theorems regarding *existence* and *generalizability*, respectively. **Existence** means that we can linearly transform the VAE embeddings of an origin and its translation such that their distance is close enough. **Generalizability** means that the linear transformation trained on the images generated by one diffusion model can be effectively applied to another different diffusion model. Finally, we show how to train the required linear transformation in practice.

### 4.1 EXISTENCE

**Theorem 1.** *Consider a well-trained diffusion model $\mathcal{F}_1$ with an encoder $\mathcal{E}_1$ from its VAE and its text-guided image-to-image functionability achieved by denoising noised images. There exists a linear transformation matrix $\mathbf{W}$, for any original image o and its translation $g_1$, such that:*

$$\mathcal{E}_1(g_1) \cdot \mathbf{W} = \mathcal{E}_1(o) \cdot \mathbf{W}. \tag{1}$$

Note that we omit the flattening operation that transforms a multi-dimensional matrix, $\mathcal{E}_1(g_1)$ or $\mathcal{E}_1(o)$, into a one-dimensional vector.

*Proof.* The proof of Theorem 1 is based on the below lemmas. Please refer to Appendix A for the proofs of lemmas. We prove the Theorem 1 here.

**Lemma 1.** *Consider the diffusion model as defined in **Theorem 1**. Define $\bar{\alpha}_t$ as the key coefficient regulating the noise level. Let $\epsilon$ denote the noise vector introduced during the diffusion process, and let $\epsilon_\theta(\mathbf{z}_t, t, \mathbf{c})$ represent the noise estimated by the diffusion model, where: $\theta$ denotes the parameters of the model, $\mathbf{z}_t$ represents the state of the system at time $t$, and $\mathbf{c}$ encapsulates the text-conditioning information. Under these conditions, the following identity holds:*

$$\mathcal{E}_1(g_1) - \mathcal{E}_1(o) = \frac{\sqrt{1 - \bar{\alpha}_t}}{\sqrt{\bar{\alpha}_t}} \left( \epsilon - \epsilon_\theta(\mathbf{z}_t, t, \mathbf{c}) \right). \tag{2}$$

**Lemma 2.** *Consider the equation $\mathbf{AX} = \mathbf{0}$, where $\mathbf{A}$ is a matrix. If $\mathbf{A}$ approximately equals to zero matrix, i.e., $\mathbf{A} \approx \mathbf{O}$, then there exists an approximate full-rank solution to the equation.*

Because a well-trained diffusion model learns robust features and associations from diverse data, it generalizes well to inference prompts that are semantically similar to the training prompts. Moreover, the inference prompts here are generated by GPT-4o based on its understanding of the images, thus sharing semantic overlap with the training prompts. As a result, the estimated noise $\epsilon_\theta(\mathbf{z}_t, t, \mathbf{c})$ closely approximates the true noise $\epsilon$. This means the difference between them is approximately equals to zero, *i.e.*, $\epsilon - \epsilon_\theta(\mathbf{z}_t, t, \mathbf{c}) \approx \mathbf{0}$. According to **Lemma 1**, this results in $\mathcal{E}_1(g_1) - \mathcal{E}_1(o) \approx \mathbf{0}$. Denote $\mathbf{T}_1$ as the matrix, in which each column is $\mathcal{E}_1(g_1) - \mathcal{E}_1(o)$ from a training pair. According to **Lemma 2** and $\mathbf{T}_1 \approx \mathbf{O}$, we have $\mathbf{T}_1 \mathbf{X} = \mathbf{0}$ has an approximate full-rank solution. That means the matrix $\mathbf{W}$ satisfying Eq. 1 exists. $\square$

**Note:** here we do **not** show that $\mathcal{E}_1(g_1) = \mathcal{E}_1(o)$ (in this case, there would be no need of $\mathbf{W}$); instead, we prove that there exists a $\mathbf{W}$ that can further minimize the distance between $\mathcal{E}_1(g_1)$ and $\mathcal{E}_1(o)$, despite the distance already being small. Please see Table 4 and Fig. 7 for experimental evidences.

### 4.2 GENERALIZABILITY

**Theorem 2.** *Following **Theorem 1**, consider a different well-trained diffusion model $\mathcal{F}_2$ and its text-guided image-to-image functionability achieved by denoising noised images. The matrix $\mathbf{W}$ can be generalized such that for any original image o and its translation $g_2$, we have:*

$$\mathcal{E}_1(g_2) \cdot \mathbf{W} = \mathcal{E}_1(o) \cdot \mathbf{W}. \tag{3}$$

*Figure 4:* The implementation of learning theoretical-expected matrix $\mathbf{W}$. Specifically, in practice, we use gradient descent to optimize a metric loss function in order to learn $\mathbf{W}$.

*Proof.* The proof of Theorem 2 is based on the below observation and lemmas. Please refer to Appendix A for the proofs of lemmas. We prove the Theorem 2 here.

**Observation 1.** *Consider two distinct matrices, $\mathbf{W}_1$ and $\mathbf{W}_2$, satisfying Eq. 1 and Eq. 3, respectively. Let $\mathbf{v}_i$ denote the vector of all singular values of $\mathbf{W}_i$, where $i \in \{1, 2\}$. Specifically, define $\mathbf{v}_i = (\sigma_i^1, \sigma_i^2, \ldots, \sigma_i^k)$, with each $\sigma_i^j$ representing an singular value of $\mathbf{W}_i$. Despite the inequality $\mathbf{W}_1 \neq \mathbf{W}_2$, as shown in Table 1, it is observed that:*

$$cos\,(\varphi) = \frac{\mathbf{v}_1 \cdot \mathbf{v}_2}{\|\mathbf{v}_1\|\|\mathbf{v}_2\|} \to 1. \tag{4}$$

*Table 1:* The $cos\,(\varphi)$ gained by compared Stable Diffusion 2 against different diffusion models. The experiments are repeated for **ten** times to calculate mean and standard deviation.

| $cos\,(\varphi)$ | SDXL | OpenDalle | ColorfulXL | Kandinsky-3 | SD3 | Kolors |
|---|---|---|---|---|---|---|
| SD2 | 0.995790 ± 0.000037 | 0.996532 ± 0.000016 | 0.998436 ± 0.000015 | 0.999788 ± 0.000009 | 0.993256 ± 0.000035 | 0.991808 ± 0.000042 |

**Lemma 3 (Singular Value Decomposition).** *Any matrix $\mathbf{A}$ can be decomposed into the product of three matrices: $\mathbf{A} = \mathbf{U}\boldsymbol{\Sigma}\mathbf{V}^*$, where $\mathbf{U}$ and $\mathbf{V}$ are orthogonal matrices, $\boldsymbol{\Sigma}$ is a diagonal matrix with non-negative singular values of $\mathbf{A}$ on the diagonal, and $\mathbf{V}^*$ is the conjugate transpose of $\mathbf{V}$.*

**Lemma 4.** *A matrix $\mathbf{A}$ has a left inverse if and only if it has full rank.*

Consider $\mathbf{T_1}$ in the proof of **Theorem 1**, and denote $\mathbf{T_2}$ as the matrix, in which each column is $\mathcal{E}_1(g_2) - \mathcal{E}_1(o)$ from a training pair. Therefore, we have $\mathbf{T}_1\mathbf{W}_1 = \mathbf{0}$ and $\mathbf{T}_2\mathbf{W}_2 = \mathbf{0}$. To prove **Theorem 2**, we only need to prove $\mathbf{T}_2\mathbf{W}_1 = \mathbf{0}$. According to **Lemma 3**, there exists orthogonal matrices, $\mathbf{U}_1, \mathbf{U}_2, \mathbf{V}_1,$ and $\mathbf{V}_2$, with diagonal matrices, $\boldsymbol{\Sigma}_1$ and $\boldsymbol{\Sigma}_2$, satisfying $\mathbf{W}_1 = \mathbf{U}_1\boldsymbol{\Sigma}_1\mathbf{V}_1^*$ and $\mathbf{W}_2 = \mathbf{U}_2\boldsymbol{\Sigma}_2\mathbf{V}_2^*$. According to **Observation 1**, there exists $\alpha > 0$ such that $\boldsymbol{\Sigma}_1 = \alpha \cdot \boldsymbol{\Sigma}_2$. Therefore, we have:

$$\mathbf{W}_1 = \mathbf{U}_1\boldsymbol{\Sigma}_1\mathbf{V}_1^* = \alpha\mathbf{U}_1\boldsymbol{\Sigma}_2\mathbf{V}_1^* = \alpha\mathbf{U}_1\left(\mathbf{U}_2^*\mathbf{W}_2\mathbf{V}_2\right)\mathbf{V}_1^* = \alpha\left(\mathbf{U}_1\mathbf{U}_2^*\right)\mathbf{W}_2\left(\mathbf{V}_2\mathbf{V}_1^*\right). \tag{5}$$

Let $\mathbf{U}_3 = \mathbf{U}_1\mathbf{U}_2^*$ and $\mathbf{V}_3 = \mathbf{V}_2\mathbf{V}_1^*$, where $\mathbf{U}_3$ and $\mathbf{V}_3$ are thus orthogonal matrices. Therefore:

$$\begin{aligned}\| \mathbf{T}_2\mathbf{W}_1 \| &= \alpha \| \mathbf{T}_2\left(\mathbf{U}_1\mathbf{U}_2^*\right)\mathbf{W}_2\left(\mathbf{V}_2\mathbf{V}_1^*\right) \| = \alpha \| \mathbf{T}_2\mathbf{U}_3\mathbf{W}_2\mathbf{V}_3 \| \\ &\leqslant \alpha \| \mathbf{T}_2\mathbf{U}_3\mathbf{W}_2 \| \cdot \| \mathbf{V}_3 \| = \alpha \| \mathbf{T}_2\mathbf{U}_3\mathbf{W}_2 \| .\end{aligned} \tag{6}$$

According to **Lemma 2 and 4**, there exists a matrix $\mathbf{K}$, such that $\mathbf{K}\mathbf{W}_2 = \mathbf{I}$. That means there exists $\mathbf{M}$, such that $\mathbf{U}_3\mathbf{W}_2 = \mathbf{W}_2\mathbf{M}$. This results in:

$$\|\mathbf{T}_2\mathbf{W}_1\| \leq \alpha\|\mathbf{T}_2\mathbf{U}_3\mathbf{W}_2\| = \alpha\|\mathbf{T}_2\mathbf{W}_2\mathbf{M}\| \leq \alpha\|\mathbf{T}_2\mathbf{W}_2\| \cdot \|\mathbf{M}\| \tag{7}$$

Considering $\mathbf{T}_2\mathbf{W}_2 = \mathbf{0}$, we have $\mathbf{T}_2\mathbf{W}_1 = \mathbf{0}$. □

### 4.3 IMPLEMENTATION

As illustrated in Fig. 4, we show how to learn the theoretical-expected matrix $\mathbf{W}$ in practice. Consider a triplet $(g, o, n)$, where $g$ is the generated image, $o$ is the origin used to generate $g$, and $n$ is a negative sample relative to $g$. We have:

$$\mathbf{z} = \mathcal{E}(g), \mathbf{z}_o = \mathcal{E}(o), \text{ and, } \mathbf{z}_n = \mathcal{E}(n), \tag{8}$$

where $\mathcal{E}$ is the encoder of VAE. Therefore, the final loss is defined as:

$$\mathcal{L} = \mathcal{L}_{mtr}(\mathbf{z} \cdot \mathbf{W}, \mathbf{z}_o \cdot \mathbf{W}, \mathbf{z}_n \cdot \mathbf{W}), \tag{9}$$

where $\mathcal{L}_{mtr}$ is a metric learning loss function that aims to bring positive data points closer together in the embedding space while pushing negative data points further apart. We use CosFace (Wang et al., 2018) here as $\mathcal{L}_{mtr}$ for its simplicity and effectiveness. Using gradient descent, we can optimize the loss function $\mathcal{L}$ to obtain the theoretically expected matrix $\mathbf{W}$.

## 5 EXPERIMENTS

### 5.1 EVALUATION PROTOCOLS AND TRAINING DETAILS

**Evaluation protocols.** We adopt two commonly used evaluation metrics for our $\text{ID}^2$ task: *i.e.*, Mean Average Precision (mAP) and Top-1 Accuracy (Acc). mAP evaluates a model's precision at *various* recall levels, while Acc measures the proportion of instances where the model's *top* prediction exactly matches the original image. Acc is stricter as it only counts when the first guess is correct.

**Training details.** We distribute the optimization of the theoretically expected matrix $\mathbf{W}$ across 8 NVIDIA A100 GPUs using PyTorch (Paszke et al., 2019). The images are resized to a resolution of $256 \times 256$ before being embedded by the VAE encoder. The peak learning rate is set to $3.5 \times 10^{-4}$, and the Adam optimizer (Kingma, 2014) is used.

### 5.2 THE CHALLENGE FROM $\text{ID}^2$

This section benchmarks popular public deep embedding models on the OriPID test dataset. As shown in Table 2 and Fig. 5, we extensively experiment on *supervised pre-trained models*, *self-supervised learning models*, *vision-language models*, and *image copy detection models*. We use these models as feature extractors, matching query features against references. The mAP and Acc are calculated by averaging the results of 7 diffusion models. Please refer to Table 7 in Appendix for the complete results. We observe that: **(1)** All existing methods **fail** on the OriPID test dataset, highlighting the importance of constructing specialized training datasets and developing new methods. Specifically, *supervised pre-trained models* overly focus on category-level similarity and thus achieve a maximum mAP of 6.2%; *self-supervised learning models* handle only subtle changes and thus achieve a maximum mAP of 11.6%; *vision-language models* return matches with overall semantic consistency, achieving a maximum mAP of 8.3%; and *image copy detection models* are trained with translation patterns different from those of the $\text{ID}^2$ task, thus achieving a maximum mAP of 29.1%. **(2)** AnyPattern (Wang et al., 2024a) achieves significantly higher mAP (29.1%) and accuracy (25.7%) compared to other existing methods. This is reasonable because AnyPattern is specifically designed for pattern generalization. Although the translation patterns generated by diffusion models in our $\text{ID}^2$ differ from the manually designed patterns in AnyPattern, there remains some generalizability.

### 5.3 VAE DIFFERS BETWEEN SEEN AND UNSEEN DIFFUSION MODELS

A common *misunderstanding* is that the generalizability of our method comes from different diffusion models sharing the same or similar VAE. In Table 3, we demonstrate that the VAE encoders used in our method **differ** between the diffusion models for generating training and testing images: **(1)** *The parameters of VAE encoders are different.* For instance, the cosine similarity of the last convolutional layer weights of the VAE encoder between Stable Diffusion 2 and Stable Diffusion XL is only 0.169. Furthermore, the number of channels in the last convolutional layer differs between Stable Diffusion 2 and Stable Diffusion 3. **(2)** *The embeddings encoded by VAEs from different diffusion models vary.* For instance, the average cosine similarity of VAE embeddings for 100,000 original images between Stable Diffusion 2 and Kandinsky-3 is close to 0. Additionally, the dimension of the VAE embedding for Stable Diffusion 2 is $4,096$, whereas for Stable Diffusion 3, it is $16,384$.

*Table 2:* Publicly available models **fail** on the test set of OriPID.

| | Method | Venue | mAP | Acc |
|---|---|---|---|---|
| Supervised Pre-trained Models | Swin-B (Liu et al., 2021) | ICCV | 3.9 | 2.7 |
| | ResNet-50 (He et al., 2016) | CVPR | 4.5 | 3.0 |
| | ConvNeXt (Liu et al., 2022) | CVPR | 4.5 | 3.1 |
| | EfficientNet (Tan & Le, 2019) | ICML | 4.6 | 3.3 |
| | ViT-B (Dosovitskiy et al., 2021) | ICLR | 6.2 | 4.6 |
| Self-supervised Learning Models | SimSiam (Chen & He, 2021) | CVPR | 1.8 | 1.0 |
| | MoCov3 (He et al., 2020) | CVPR | 2.1 | 1.2 |
| | DINOv2 (Oquab et al., 2023) | TMLR | 4.3 | 2.9 |
| | MAE (He et al., 2022) | CVPR | 11.6 | 9.2 |
| | SimCLR (Chen et al., 2020) | ICML | 11.3 | 9.7 |
| Vision-language Models | CLIP (Radford et al., 2021) | ICML | 2.9 | 1.8 |
| | SLIP (Mu et al., 2022) | ECCV | 5.4 | 3.7 |
| | ZeroVL (Cui et al., 2022) | ECCV | 5.6 | 3.8 |
| | BLIP (Li et al., 2022) | ICML | 8.3 | 5.9 |
| Image Copy Detection Models | ASL (Wang et al., 2023a) | AAAI | 5.2 | 4.1 |
| | CNNCL (Yokoo, 2021) | PMLR | 6.3 | 5.0 |
| | BoT (Wang et al., 2021) | PMLR | 10.5 | 8.2 |
| | SSCD (Pizzi et al., 2022) | CVPR | 14.8 | 12.5 |
| | AnyPattern (Wang et al., 2024a) | Arxiv | 29.1 | 25.7 |

*Table 3:* VAE **differs** between seen and unseen diffusion models.

| Cosine Sim. | | SDXL | OpDa | CoXL | Kan3 | SD3 | Kolor |
|---|---|---|---|---|---|---|---|
| SD2 | Conv. | 0.169 | 0.169 | 0.169 | 0.002 | - | 0.169 |
| | Embed. | 0.120 | 0.121 | 0.120 | 0.023 | - | 0.120 |

**Generation** **Matching**

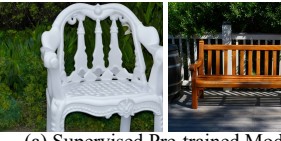

(a) Supervised Pre-trained Models

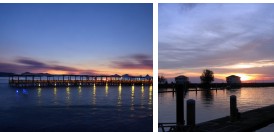

(b) Self-supervised Learning Models

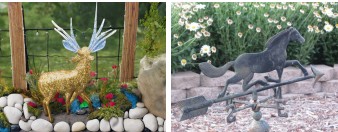

(c) Vision-language Models

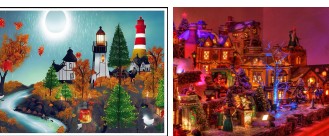

(d) Image Copy Detection Models

*Figure 5:* Examples of **failure** cases for each kind of model.

## 5.4 THE EFFECTIVENESS OF OUR METHOD

This section demonstrates the effectiveness of our method in terms of (1) *generalizability*, (2) *efficiency*, (3) *robustness*, and (4) the *consistency* between theory and experiments. The experimental results for 'Unseen' are obtained by averaging the results from six different unseen diffusion models.

**Our method is much more generalizable than others.** In Table 4, we compare our method with common similarity-based methods (incorporating domain generalization designs), all trained on the OriPID training dataset. The mAP and Acc for 'Unseen' are calculated by averaging the results of 6 unseen diffusion models. Please refer to Table 8 and Section E in the Appendix for the complete results and failure cases, respectively. We make three observations: **(1)** On **unseen** data, our method demonstrates significant performance superiority over common similarity-based models. Specifically, compared against the best one, we achieve a superiority of $+31.6\%$ mAP and $+35.1\%$ Acc. **(2)** Although domain generalization methods alleviate the generalization problem, they are still not satisfactory compared to ours (with at least a $-10.8\%$ mAP and $-9.1\%$ Acc). Moreover, those with the best performance suffer from severe efficiency issues, as detailed in the next section. **(3)** On the **seen** data, we achieve comparable performance with others. Specifically, there is a $0.2\%$ mAP and $1.5\%$ Acc superiority compared to the best one.

**Our method outperforms others in terms of efficiency.** Efficiency is crucial for the proposed task, as it often involves matching a query against a large-scale database in real-world scenarios. In Table 4, we compare the efficiency of our method with others regarding (1) *training*, (2) *feature extraction*, and (3) *matching*. We draw three observations: **(1) Training:** Learning a matrix based on VAE embeddings is more efficient compared to training deep models on raw images. Specifically, our method is 8.6 times faster than the nearest competitor. **(2) Feature extraction:** Compared to other models that use deep networks, such as ViT (Dosovitskiy et al., 2021), the VAE encoder we use is relatively lightweight, resulting in faster feature extraction. **(3) Matching:** Compared to the best domain generalization models, QAConv-GS (Liao et al., 2022), which use feature *maps* for matching, our method still relies on feature *vectors*. This leads to an $\textbf{875}\times$ superiority in matching speed.

**Our method is relatively robust against different attacks.** In the real world, the quality of an image may deteriorate during transmission. As shown in Fig. 6, we apply varying intensities of Gaussian blur and JPEG compression, following previous works such as (Wang et al., 2023b; Chen et al., 2024), to evaluate the robustness of our method. It is observed that the side effects of these

*Table 4:* Our method excels in **performance** while keeping **efficiency**. 'mAP' and 'Acc' are in percentage; 'Train', 'Extract', and 'Match' are in 'h', '$10^{-4}$ s/img', and '$10^{-10}$ s/pair', respectively.

| | Method | Venue | Seen ↑ | | Unseen ↑ | | Efficiency ↓ | | |
|---|---|---|---|---|---|---|---|---|---|
| | | | mAP | Acc | mAP | Acc | Train | Extract | Match |
| Similarity -based Models | Circle loss (Sun et al., 2020) | CVPR | 70.4 | 64.3 | 53.9 | 48.5 | 1.79 | 2.81 | 0.80 |
| | SoftMax (LeCun et al., 1989) | NC | 82.7 | 78.3 | 55.0 | 49.4 | 2.25 | 2.81 | 0.80 |
| | CosFace (Wang et al., 2018) | CVPR | 87.1 | 83.2 | 52.2 | 46.5 | 2.43 | 2.81 | 0.80 |
| General- izable Models | IBN-Net (Pan et al., 2018) | ECCV | 88.6 | 85.1 | 54.6 | 49.0 | 2.03 | 3.42 | 2.14 |
| | TransMatcher (Liao et al., 2021) | NIPS | 65.6 | 60.3 | 65.3 | 60.7 | 1.84 | 2.30 | 941 |
| | QAConv-GS (Liao et al., 2022) | CVPR | 78.8 | 74.9 | 75.8 | 72.3 | 1.47 | 2.30 | 464 |
| **Ours** | Embeddings of VAE | - | 51.0 | 47.0 | 46.9 | 43.0 | - | 1.59 | 4.25 |
| | With Linear Transformation | - | **88.8** | **86.6** | **86.6** | **84.5** | **0.17** | **1.59** | **0.53** |
| | Upper: Train&Test Same Domain | - | 88.8 | 86.6 | 92.0 | 90.4 | 0.17 | 1.59 | 0.53 |

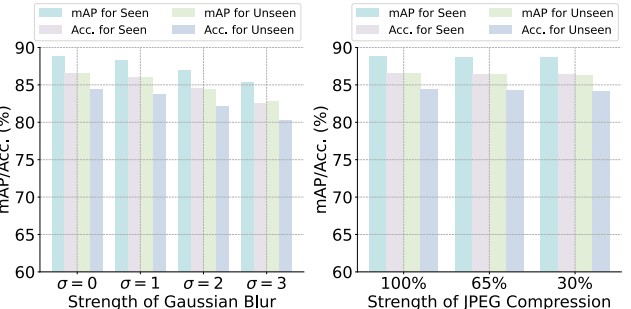
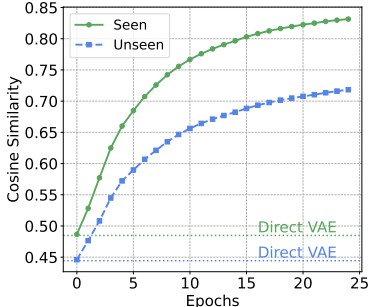

*Figure 6:* Our method demonstrates a certain level of **robustness** against different *types* and *intensities* of attacks.

*Figure 7:* As expected, the cosine similarities **increase** w.r.t. epochs.

attacks are relatively minor. For instance, for the unseen diffusion models, the strongest Gaussian blur ($\sigma = 3$) reduces the mAP by only 3.7%, while the strongest compression (30%) decreases the mAP by just 0.3%. It is important to note that our models are **not** trained with these attacks.

**Our training scheme successfully learns the theory-anticipated matrix W.** In the theorems, we have proven that $\mathcal{E}_1(g_1/g_2) \cdot \mathbf{W} = \mathcal{E}_1(o) \cdot \mathbf{W}$ holds ideally. In Fig. 7, we experimentally show this phenomenon. Specifically, we first calculate two cosine similarities of $< \mathcal{E}_1(g_1) \cdot \mathbf{W}, \mathcal{E}_1(o) \cdot \mathbf{W} >$ (seen) and $< \mathcal{E}_1(g_2) \cdot \mathbf{W}, \mathcal{E}_1(o) \cdot \mathbf{W} >$ (unseen), and then plot their changes with respect to the epochs. We observe that: **(1)** as expected, the two cosine similarities increase during training; and **(2)** the cosine similarities of the seen diffusion models are higher than those of the unseen ones, which is reasonable due to a certain degree of overfitting.

## 5.5 ABLATION STUDY

In this section, we ablate the proposed method by (1) using different VAE *encoders*, (2) supervising the training with different *loss* functions, (3) exploring the minimum *rank* of **W**, and (4) experimentally exploring *beyond* the theoretical guarantees.

**Our method is insensitive to the choice of VAE encoder.** In Table 5, we replace the VAE encoder from Stable Diffusion 2 with two different encoders from Open-Sora (Zheng et al., 2024) and Open-Sora-Plan (PKU-Yuan & etc., 2024). It is observed that, despite using significantly different well-trained VAEs, such as ones for videos, the performance drop is minimal (less than 1%). This observation experimentally extends the Eq. 1 from $\mathcal{E}_1(g_1) \cdot \mathbf{W} = \mathcal{E}_1(o) \cdot \mathbf{W}$ to $\mathcal{E}_2(g_1) \cdot \mathbf{W} = \mathcal{E}_2(o) \cdot \mathbf{W}$, where $\mathcal{E}_2$ is an encoder from a totally different VAE.

**In practice, selecting an appropriate supervision for learning W is essential.** In Table 6, we replace the used supervision CosFace (Wang et al., 2018) with two weaker supervisions, *i.e.*, SoftMax (LeCun et al., 1989) and Circle loss (Sun et al., 2020). We observe that switching to Circle loss results in a drop in mAP for seen and unseen categories by 3.9% and 4.1%, respectively. Furthermore, using SoftMax leads to mAP drops of 12.7% and 24.2% for the two categories, respectively. We infer this is because: while our theorems guarantee the distance between translation and its origin, many negative samples serve as distractors during retrieval. Without appropriate hard negative solutions, these distractors compromise the final performance.

Table 5: Ablation for choices of VAE encoders.

| VAE | Seen ↑ | | Unseen ↑ | |
|---|---|---|---|---|
| | mAP | Acc | mAP | Acc |
| Open-Sora | 86.3 | 83.5 | 86.5 | 84.2 |
| Open-Sora-Plan | 88.8 | 86.4 | 86.1 | 84.0 |
| Stable Diffusion 2 | **88.8** | **86.6** | **86.6** | **84.5** |

Table 6: Ablation for different supervision losses.

| Supervision | Seen ↑ | | Unseen ↑ | |
|---|---|---|---|---|
| | mAP | Acc | mAP | Acc |
| SoftMax | 76.1 | 72.6 | 62.4 | 59.0 |
| Circle loss | 84.9 | 82.0 | 82.5 | 80.4 |
| CosFace | **88.8** | **86.6** | **86.6** | **84.5** |

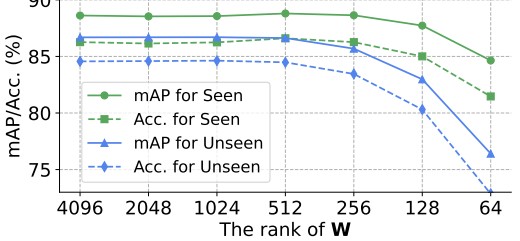

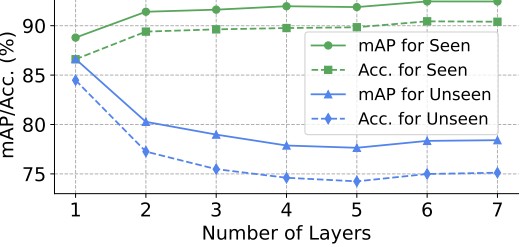

Figure 8: The change in performance with respect to the rank of $\mathbf{W}$.

Figure 9: The change in performance with respect to the number of layers.

**To improve efficiency, the rank of W can be relatively low.** Assume the matrix $\mathbf{W}$ has a shape of $n \times m$, where $n$ is the dimension of the VAE embedding and $m$ is a hyperparameter. We show that $\mathbf{W}$ is approximately full-rank in the proof of *existence*, and expect that $m \leq n$ in the proof of *generalization*. Therefore, the rank of $\mathbf{W}$ is $m$. Experimentally, $n = 4,096$, and we explore the minimum rank of $\mathbf{W}$ from $4,096$ as shown in Fig. 8. It is observed that: **(1)** From $4,096$ to $512$, the performance remains nearly unchanged. This suggests that we can train a relatively low-rank $\mathbf{W}$ to improve efficiency in real-world applications. **(2)** It is expected to see a performance decrease when reducing the rank from $512$ to $64$. This is because a matrix with too low rank cannot carry enough information to effectively linearly transform the VAE embeddings.

**Using a multilayer perceptron (MLP) with activation function instead of the theoretically expected W leads to overfitting.** In the theoretical section, we proved the existence and generalization of $\mathbf{W}$ using concepts from *diffusion models* and *linear algebra*. A natural experimental extension of this is to use an MLP with activation functions to replace the simple linear transformation ($\mathbf{W}$). Although linear algebra theory cannot guarantee these cases, we can still explore them experimentally. Experimentally, we increase the number of layers from 1 to 7, all using ReLU activation and residual connections. As shown in Fig. 9, we observe overfitting in one type of diffusion model. Specifically, on one hand, the performance on seen diffusion models improves. For example, with 2 layers, the mAP increases to $91.4\%$ ($+2.6\%$), and Acc rises to $89.4\%$ ($+2.8\%$). However, on the other hand, a significant performance drop is observed on unseen diffusion models: with 2 layers, the mAP decreases from $86.6\%$ to $80.3\%$ ($-6.3\%$), and Acc drops from $84.5\%$ to $77.3\%$ ($-7.2\%$). The performance drop problem becomes even more severe when using more layers.

## 6 CONCLUSION

This paper explores popular text-guided image-to-image diffusion models from a novel perspective: retrieving the original image of a query translated by these models. The proposed task, ID², is both important and timely, especially as awareness of security concerns posed by diffusion models grows. To support this task, we introduce the first ID² dataset, OriPID, designed with a focus on addressing generalization challenges. Specifically, the training set is generated by one diffusion model, while the test set is generated by seven different models. Furthermore, we propose a simple, generalizable solution with theoretical guarantees: First, we theoretically prove the existence of linear transformations that minimize the distance between the VAE embeddings of a query and its original image. Then, we demonstrate that the learned linear transformations generalize across different diffusion models, *i.e.*, the VAE encoder and the learned transformations can effectively embed images generated by new diffusion models.

**Limitation.** We note that certain methods, such as InstructPix2Pix (Brooks et al., 2023) and IP-Adapter (Ye et al., 2023), perform text-guided image-to-image tasks in paradigms that go beyond the scope of our theorems. For a more detailed discussion, please refer to the Appendix (Section F).

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

## A    PROOFS OF LEMMAS

> **Lemma 1.** *Consider the diffusion model as defined in **Theorem 1**. Define $\bar{\alpha}_t$ as the key coefficient regulating the noise level. Let $\epsilon$ denote the noise vector introduced during the diffusion process, and let $\epsilon_\theta(\mathbf{z}_t, t, \mathbf{c})$ represent the noise estimated by the diffusion model, where: $\theta$ denotes the parameters of the model, $\mathbf{z}_t$ represents the state of the system at time $t$, and $\mathbf{c}$ encapsulates the text-conditioning information. Under these conditions, the following identity holds:*
>
> $$\mathcal{E}_1(g_1) - \mathcal{E}_1(o) = \frac{\sqrt{1 - \bar{\alpha}_t}}{\sqrt{\bar{\alpha}_t}} (\epsilon - \epsilon_\theta(\mathbf{z}_t, t, \mathbf{c})). \tag{10}$$

*Proof.* Denote $\mathbf{z}_0 = \mathcal{E}_1(o)$ and $\mathbf{z}'_0$ as $\mathbf{z}_0$ after adding noise and denoising. Therefore, we have

$$\mathcal{E}_1(g_1) - \mathcal{E}_1(o) = \mathcal{E}_1(\mathcal{D}_1(\mathbf{z}'_0)) - \mathbf{z}_0 = \mathbf{z}'_0 - \mathbf{z}_0, \tag{11}$$

where $\mathcal{D}_1$ is the decoder of VAE.

Given an initial data point $\mathbf{z}_0$, the forward process in a diffusion model adds noise to the data step by step. The expression for $\mathbf{z}_t$ at a specific timestep $t$ can be written as:

$$\mathbf{z}_t = \sqrt{\bar{\alpha}_t}\mathbf{z}_0 + \sqrt{1 - \bar{\alpha}_t}\epsilon. \tag{12}$$

To denoise $\mathbf{z}_t$ and recover an estimate of the original data $\mathbf{z}_0$, the reverse process is used. A neural network $\theta$ is trained to predict the noise $\epsilon$ added to $\mathbf{z}_0$. The denoised data $\mathbf{z}'_0$ can be expressed as:

$$\mathbf{z}'_0 = \frac{1}{\sqrt{\bar{\alpha}_t}} \left( \mathbf{z}_t - \sqrt{1 - \bar{\alpha}_t}\epsilon_\theta(\mathbf{z}_t, t, \mathbf{c}) \right). \tag{13}$$

Therefore, we have:

$$\begin{aligned}
\mathcal{E}_1(g_1) - \mathcal{E}_1(o) &= \mathbf{z}'_0 - \mathbf{z}_0 \\
&= \frac{1}{\sqrt{\bar{\alpha}_t}} \left( \mathbf{z}_t - \sqrt{1 - \bar{\alpha}_t}\epsilon_\theta(\mathbf{z}_t, t, \mathbf{c}) \right) - \mathbf{z}_0 \\
&= \frac{1}{\sqrt{\bar{\alpha}_t}} \left( \sqrt{\bar{\alpha}_t}\mathbf{z}_0 + \sqrt{1 - \bar{\alpha}_t}\epsilon - \sqrt{1 - \bar{\alpha}_t}\epsilon_\theta(\mathbf{z}_t, t, \mathbf{c}) \right) - \mathbf{z}_0 \\
&= \frac{\sqrt{1 - \bar{\alpha}_t}}{\sqrt{\bar{\alpha}_t}} (\epsilon - \epsilon_\theta(\mathbf{z}_t, t, \mathbf{c})).
\end{aligned} \tag{14}$$

The Eq. 10 is proved. □

> **Lemma 2.** *Consider the equation $\mathbf{AX} = \mathbf{0}$, where $\mathbf{A}$ is a matrix. If $\mathbf{A}$ approximately equals to zero matrix, i.e., $\mathbf{A} \approx \mathbf{O}$, then there exists an approximate full-rank solution to the equation.*

*Proof.* Consider a matrix $\mathbf{A} \in \mathbb{R}^{m \times n}$. According to **Lemma 3**, there exists orthogonal matrices $\mathbf{U} \in \mathbb{R}^{m \times m}$ and $\mathbf{V} \in \mathbb{R}^{n \times n}$, and diagonal matrix $\mathbf{\Sigma} \in \mathbb{R}^{m \times n}$ with non-negative singular values, such that, $\mathbf{A} = \mathbf{U\Sigma V}^*$. Therefore, the linear equation can be transformed as:

$$\mathbf{U\Sigma V}^*\mathbf{X} = \mathbf{0}. \tag{15}$$

Considering $\mathbf{U}^*\mathbf{U} = \mathbf{I}$ and denoting $\mathbf{X}' = \mathbf{V}^*\mathbf{X}$, we have $\mathbf{\Sigma X}' = \mathbf{0}$. Because $\mathbf{A} \approx \mathbf{O}$, all of its singular values approximately equals to $0$. Considering the floating-point precision we need, $\mathbf{\Sigma X}' = \mathbf{0}$ could be regarded as:

$$\left( \begin{pmatrix} \sigma_0 & & & \\ & \sigma_1 & & \\ & & \ddots & \\ & & & \sigma_r \\ \begin{pmatrix} 0 & & & \\ & 0 & & \\ & & \ddots & \\ & & & 0 \end{pmatrix} \end{pmatrix} \begin{pmatrix} 0 & & & \\ & 0 & & \\ & & \ddots & \\ & & & 0 \\ \begin{pmatrix} 0 & & & \\ & 0 & & \\ & & \ddots & \\ & & & 0 \end{pmatrix} \end{pmatrix} \right) \mathbf{X}' = \mathbf{0}, \tag{16}$$

where $r$ is the number of non-zero singular values. Therefore, there exists an $\mathbf{Z}' \in \mathbb{R}^{n \times k}$ with rank $= \min(m, n) - r$. When $k \leqslant \min(m, n) - r$, $\mathbf{Z}'$ is full rank, *i.e*, $\mathbf{Z} = \mathbf{V}\mathbf{Z}'$ is an approximate full-rank solution to the linear equation $\mathbf{A}\mathbf{X} = \mathbf{0}$. $\qquad\square$

**Lemma 3 (Singular Value Decomposition).** *Any matrix $\mathbf{A}$ can be decomposed into the product of three matrices: $\mathbf{A} = \mathbf{U}\mathbf{\Sigma}\mathbf{V}^*$, where $\mathbf{U}$ and $\mathbf{V}$ are orthogonal matrices, $\mathbf{\Sigma}$ is a diagonal matrix with non-negative singular values of $\mathbf{A}$ on the diagonal, and $\mathbf{V}^*$ is the conjugate transpose of $\mathbf{V}$.*

*Proof.* Consider a matrix $\mathbf{A} \in \mathbb{R}^{m \times n}$. The matrix $\mathbf{A}^*\mathbf{A}$ is therefore symmetric and positive semi-definite, which means the matrix is diagonalizable with an eigendecomposition of the form:

$$\mathbf{A}^*\mathbf{A} = \mathbf{V}\Lambda\mathbf{V}^* = \sum_{i=1}^{n} \lambda_i \mathbf{v}_i \mathbf{v}_i^* = \sum_{i=1}^{n} (\sigma_i)^2 \mathbf{v}_i \mathbf{v}_i^*, \tag{17}$$

where $\mathbf{V}$ is an orthonormal matrix whose columns are the eigenvectors of $\mathbf{A}^*\mathbf{A}$.

We have defined the singular value $\sigma_i$ as the square root of the $i$-th eigenvalue; we know we can take the square root of our eigenvalues because positive semi-definite matrices can be equivalently characterized as matrices with non-negative eigenvalues.

For the $i$-th eigenvector-eigenvalue pair, we have

$$\mathbf{A}^*\mathbf{A}\mathbf{v}_i = (\sigma_i)^2 \mathbf{v}_i. \tag{18}$$

Define a new vector $\mathbf{u}_i$, such that,

$$\mathbf{u}_i = \frac{\mathbf{A}\mathbf{v}_i}{\sigma_i}. \tag{19}$$

This construction enables $\mathbf{u}_i$ as a unit eigenvector of $\mathbf{A}\mathbf{A}^*$. Now let $\mathbf{V}$ be an $n \times n$ matrix – because $\mathbf{A}\mathbf{A}^*$ is $n \times n$ – where the $i$-th column is $\mathbf{v}_i$; let $\mathbf{U}$ be an $m \times m$ matrix – because $\mathbf{A}\mathbf{v}_i$ is an $m$-vector – where the $i$-th column is $\mathbf{u}_i$; and let $\mathbf{\Sigma}$ be a diagonal matrix whose $i$-th element is $\sigma_i$. Then we can express the relationships we have so far in matrix form as:

$$\begin{aligned} \mathbf{U} &= \mathbf{A}\mathbf{V}\mathbf{\Sigma}^{-1}, \\ \mathbf{U}\mathbf{\Sigma} &= \mathbf{A}\mathbf{V}, \\ \mathbf{A} &= \mathbf{U}\mathbf{\Sigma}\mathbf{V}^*, \end{aligned} \tag{20}$$

where we use the fact that $\mathbf{V}\mathbf{V}^* = I$ and $\mathbf{\Sigma}^{-1}$ is a diagonal matrix where the $i$-th value is the reciprocal of $\sigma_i$. $\qquad\square$

**Lemma 4.** *A matrix $\mathbf{A}$ has a left inverse if and only if it has full rank.*

*Proof.* To prove Lemma 4, we must demonstrate two directions: if a matrix $\mathbf{A}$ has a left inverse, then it must have full rank, and conversely, if a matrix $\mathbf{A}$ has full rank, then it has a left inverse.

(1) Suppose $\mathbf{A} \in \mathbb{R}^{m \times n}$ has a left inverse $\mathbf{B} \in \mathbb{R}^{n \times m}$ such that $\mathbf{B}\mathbf{A} = \mathbf{I}_n$. Because the $\mathbf{I}_n$ is of rank $n$, the matrix $\mathbf{A}\mathbf{B}$ must have rank $n$. Considering the inequality:

$$n = \text{rank}(\mathbf{B}\mathbf{A}) \leq \min(\text{rank}(\mathbf{A}), \text{rank}(\mathbf{B})) \leq \text{rank}(\mathbf{A}) \leq \min(m, n) \leq n, \tag{21}$$

we have $\text{rank}(\mathbf{A}) = n$, *i.e.*, $\mathbf{A}$ has full rank.

(2) Suppose $\mathbf{A} \in \mathbb{R}^{m \times n}$ has full rank, *i.e.*, $\text{rank}(\mathbf{A}) = \min(m, n)$. We have the rows of $\mathbf{A}$ are linearly independent, and thus there exists an $n \times m$ matrix $\mathbf{C}$ such that $\mathbf{C}\mathbf{A} = \mathbf{I}_n$. That means $\mathbf{C}$ is a left inverse of $\mathbf{A}$. $\qquad\square$

# B  PROMPT AND GENERATION EXAMPLES

In Fig. 10, we present several prompts along with their corresponding generated images from our dataset, OriPID. The dataset comprehensively covers a wide range of subjects commonly found in real-world scenarios, such as (a) natural sceneries, (b) cultural architectures, (c) lively animals, (d) luxuriant plants, (e) artistic paintings, and (f) indoor items. It is important to note that in the training set, for each original image, OriPID contains 20 prompts with corresponding generated images, and for illustrative purposes, we only show 4 of them in Fig. 10.

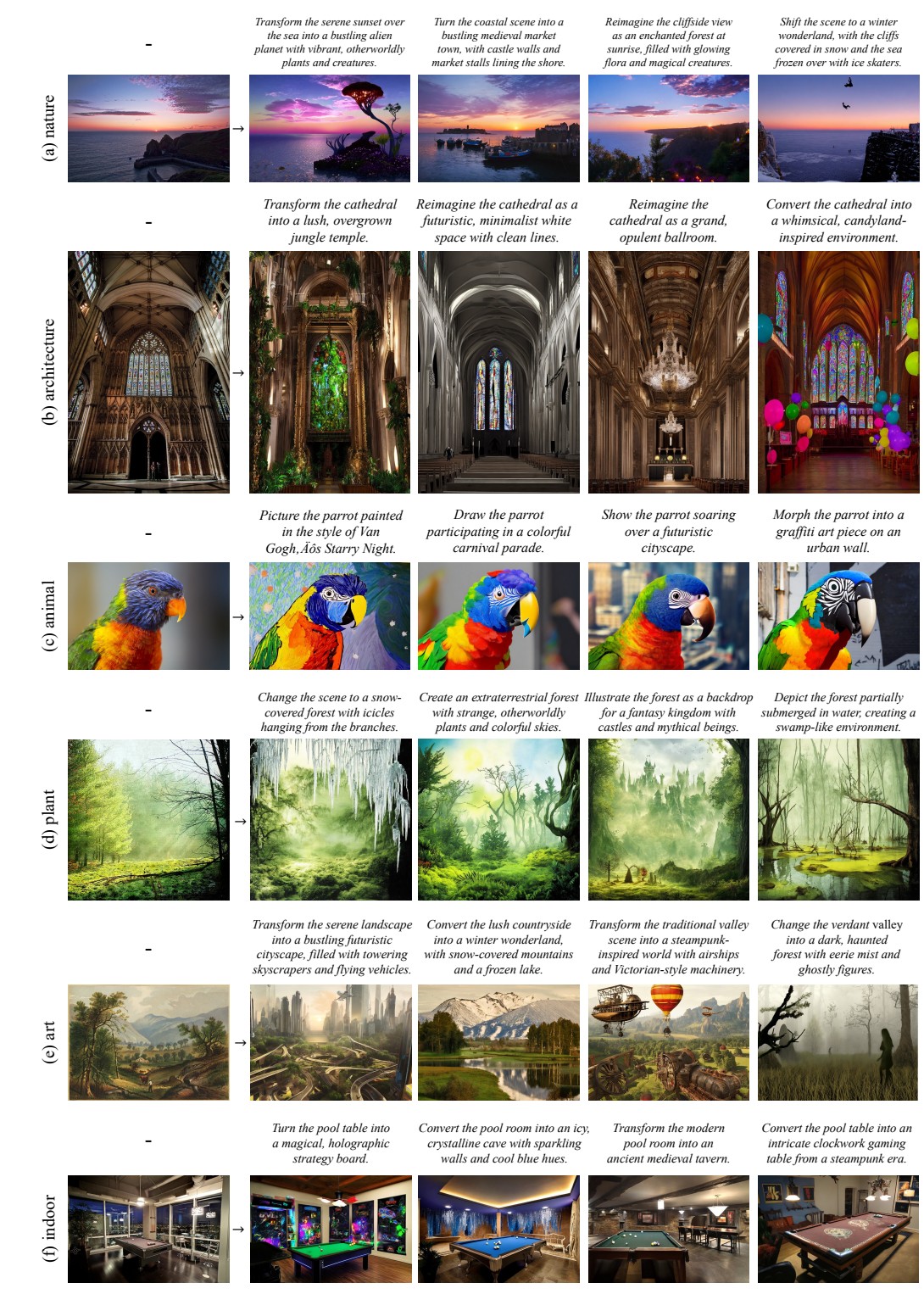

*Figure 10:* Illustration of prompts and corresponding generated images for 6 different subjects in our dataset. Our dataset comprehensively includes various subjects found in the real world.

## C   IMPLEMENTATION OF GPT-4O

As shown in Fig. 11, we request GPT-4o to generate 20 different prompts for each original image.

+ I am doing image-to-image translation. Could you think of creative prompts to translate this image to different ones? Keep them creative, and only return 20 different prompts.

*Figure 11:* The script for requesting GPT-4o to generate 20 different prompts for each original image.

*Table 7:* The performance of publicly available models on 7 different diffusion models.

| | Method | SD2 ↑ | | SDXL ↑ | | OpDa ↑ | | CoXL ↑ | | Kan3 ↑ | | SD3 ↑ | | Kolor ↑ | |
|---|---|---|---|---|---|---|---|---|---|---|---|---|---|---|---|
| | | mAP | Acc | mAP | Acc | mAP | Acc | mAP | Acc | mAP | Acc | mAP | Acc | mAP | Acc |
| | Swin-B | 3.1 | 2.0 | 2.9 | 1.9 | 4.1 | 2.9 | 4.2 | 3.1 | 6.8 | 4.7 | 2.9 | 1.9 | 3.0 | 2.0 |
| Supervised | ResNet-50 | 3.8 | 2.6 | 3.1 | 2.0 | 5.3 | 3.7 | 4.5 | 3.2 | 8.1 | 5.7 | 3.4 | 2.0 | 3.4 | 2.2 |
| Pre-trained | ConvNeXt | 3.5 | 2.1 | 3.3 | 2.2 | 4.7 | 3.3 | 5.0 | 3.5 | 8.4 | 6.2 | 3.5 | 2.4 | 3.6 | 2.6 |
| Models | EfficientNet | 2.9 | 1.9 | 2.9 | 2.0 | 4.9 | 3.4 | 5.4 | 3.9 | 8.7 | 6.5 | 3.3 | 2.2 | 4.1 | 3.0 |
| | ViT-B | 4.1 | 2.8 | 4.5 | 3.1 | 7.2 | 5.5 | 6.7 | 5.0 | 11.2 | 8.7 | 4.1 | 2.8 | 5.6 | 4.3 |
| | SimSiam | 1.5 | 1.0 | 1.2 | 0.7 | 1.8 | 0.9 | 1.7 | 1.0 | 3.1 | 1.9 | 1.5 | 0.8 | 1.4 | 0.8 |
| Self- | MoCov3 | 1.4 | 0.8 | 1.5 | 0.9 | 2.4 | 1.3 | 2.2 | 1.3 | 3.8 | 2.4 | 1.9 | 1.1 | 1.6 | 1.0 |
| supervised | DINOv2 | 2.6 | 1.6 | 2.7 | 1.7 | 4.6 | 3.0 | 5.5 | 3.6 | 8.4 | 5.9 | 2.9 | 1.9 | 3.6 | 2.6 |
| Learning | MAE | 14.9 | 11.4 | 10.0 | 8.0 | 13.1 | 10.5 | 8.1 | 6.4 | 17.6 | 14.3 | 11.2 | 8.5 | 6.5 | 5.1 |
| Models | SimCLR | 6.0 | 4.2 | 7.0 | 5.2 | 13.5 | 10.6 | 13.0 | 10.1 | 23.7 | 19.3 | 7.3 | 12.0 | 8.8 | 6.7 |
| | CLIP | 2.6 | 1.7 | 2.1 | 1.4 | 3.1 | 2.1 | 3.2 | 2.0 | 4.2 | 2.7 | 2.5 | 1.6 | 2.1 | 0.7 |
| Vision- | SLIP | 5.6 | 3.8 | 3.5 | 2.3 | 5.8 | 4.0 | 4.9 | 3.3 | 9.1 | 6.7 | 5.4 | 3.5 | 3.8 | 2.5 |
| language | ZeroVL | 5.2 | 3.5 | 4.4 | 2.9 | 6.4 | 4.4 | 4.5 | 3.2 | 9.8 | 6.9 | 4.7 | 3.0 | 4.3 | 3.0 |
| Models | BLIP | 6.8 | 4.8 | 6.5 | 4.5 | 9.9 | 7.0 | 8.7 | 6.3 | 13.8 | 10.2 | 6.0 | 3.9 | 6.4 | 4.5 |
| | ASL | 2.3 | 1.7 | 3.0 | 2.3 | 5.6 | 4.4 | 5.7 | 4.6 | 10.3 | 8.7 | 2.7 | 2.1 | 3.7 | 2.9 |
| Image Copy | CNNCL | 4.0 | 2.9 | 4.2 | 3.2 | 8.3 | 6.7 | 5.7 | 4.5 | 12.2 | 9.9 | 3.7 | 2.7 | 6.3 | 5.0 |
| Detection | BoT | 6.6 | 4.9 | 6.1 | 4.4 | 10.4 | 8.2 | 12.5 | 10.2 | 20.6 | 16.8 | 7.4 | 5.4 | 9.3 | 7.3 |
| Models | SSCD | 9.7 | 7.7 | 8.7 | 6.8 | 16.4 | 14.0 | 18.1 | 15.6 | 28.1 | 24.6 | 9.0 | 6.8 | 14.1 | 11.9 |
| | AnyPattern | 17.6 | 14.3 | 18.5 | 15.7 | 33.0 | 29.2 | 37.8 | 34.0 | 48.0 | 43.9 | 18.2 | 15.0 | 30.7 | 27.5 |

*Table 8:* The performance of our trained models on 7 different diffusion models. Note that these models are trained on images generated by SD2 and tested on images from multiple models.

| | Method | SD2 ↑ | | SDXL ↑ | | OpDa ↑ | | CoXL ↑ | | Kan3 ↑ | | SD3 ↑ | | Kolor ↑ | |
|---|---|---|---|---|---|---|---|---|---|---|---|---|---|---|---|
| | | mAP | Acc | mAP | Acc | mAP | Acc | mAP | Acc | mAP | Acc | mAP | Acc | mAP | Acc |
| Similarity | Circle loss | 70.4 | 64.3 | 56.2 | 50.1 | 56.5 | 51.8 | 41.6 | 37.0 | 60.0 | 53.8 | 65.6 | 59.3 | 43.5 | 39.2 |
| -based | SoftMax | 82.7 | 78.3 | 62.4 | 56.5 | 58.3 | 53.0 | 37.3 | 32.2 | 52.5 | 46.0 | 75.9 | 70.2 | 43.6 | 38.7 |
| Models | CosFace | 87.1 | 83.2 | 63.7 | 58.2 | 56.7 | 51.7 | 30.5 | 25.2 | 47.5 | 40.6 | 71.5 | 65.5 | 43.0 | 38.0 |
| General- | IBN-Net | 88.6 | 85.1 | 65.7 | 60.1 | 59.4 | 54.2 | 33.3 | 28.3 | 49.8 | 42.8 | 74.0 | 68.3 | 45.4 | 40.5 |
| izable | TransMatcher | 65.6 | 60.3 | 60.6 | 55.8 | 67.9 | 63.6 | 61.7 | 57.4 | 68.9 | 64.2 | 64.7 | 59.2 | 67.9 | 63.9 |
| Models | QAConv-GS | 78.8 | 74.9 | 71.6 | 67.5 | 77.4 | 74.3 | 73.6 | 70.5 | 75.2 | 71.2 | 77.3 | 73.6 | 79.5 | 76.9 |
| | VAE Embed. | 51.0 | 47.0 | 38.3 | 33.8 | 42.3 | 38.6 | 51.6 | 48.8 | 54.7 | 50.4 | 47.7 | 42.9 | 46.9 | 43.6 |
| **Ours** | Linear Trans. | **88.8** | **86.6** | **81.5** | **78.8** | **87.3** | **85.3** | **89.3** | **87.7** | **85.7** | **83.3** | **85.7** | **82.9** | **90.3** | **88.8** |
| | Upper | 88.8 | 86.6 | 84.9 | 82.4 | 90.8 | 89.2 | 93.1 | 91.9 | 95.4 | 94.3 | 93.7 | 92.0 | 94.0 | 92.8 |

## D   Complete Experiments for 7 Diffusion Models

We provide two types of complete experiments for seven different diffusion models: (1) Table 7 presents the results from directly testing publicly available models on the OriPID test dataset; and (2) Table 8 shows the results from testing models that we trained on the OriPID training dataset, which contains only images generated by Stable Diffusion 2.

## E   Failure Cases and Potential Directions

**Failure cases.** As shown in Fig. 12, we observe that our model may fail when negative samples are too visually similar to the queries. This *hard negative* problem is reasonable because our method relies on the VAE embeddings, which capture high-level representations and is insensitive to subtle changes. As a result, visually similar negative samples can produce embeddings that are close to those of the queries, leading to inaccurate matchings.

**Potential directions.** The *hard negative* problem has been studied in the Image Copy Detection (ICD) community, as exemplified by ASL (Wang et al., 2023a). It learns to assign a larger norm to the deep

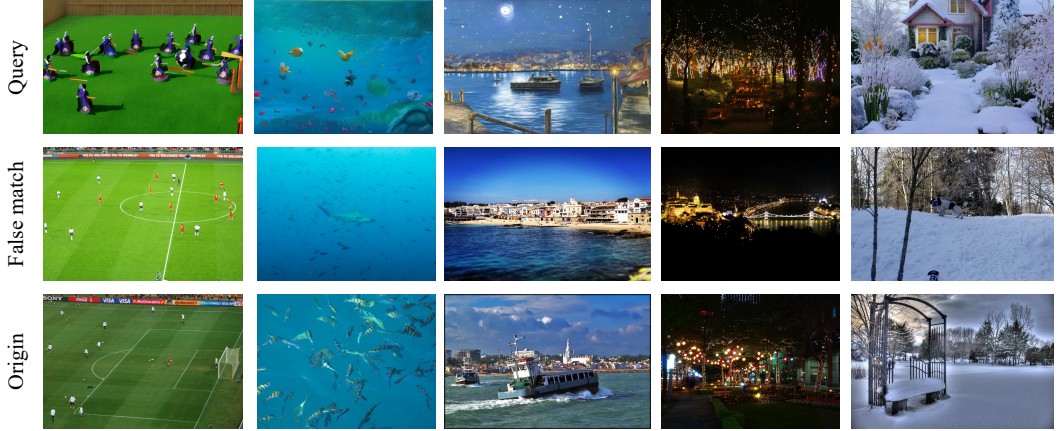

*Figure 12:* This illustration shows failure cases predicted by our method. We have identified that our model may fail when encountering *hard negative* samples.

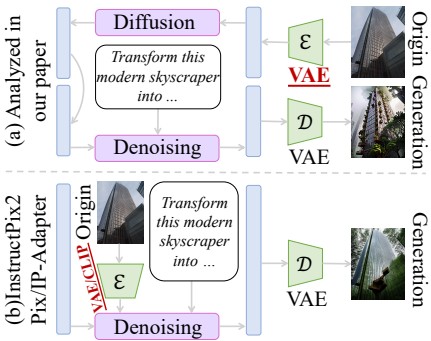

*Figure 13:* Different paradigms used by text-guided image-to-image translations.

*Table 9:* The generalization results on InstructPix2Pix (Brooks et al., 2023) and IP-Adapter (Ye et al., 2023). No method succeeds in generalizing to IP-Adapter.

| | Method | InstructP2P ↑ | | IP-Adapter ↑ | |
|---|---|---|---|---|---|
| | | mAP | Acc | mAP | Acc |
| Similarity -based Models | Circle loss | 44.9 | 42.7 | 6.2 | 4.0 |
| | SoftMax | 21.5 | 19.2 | 5.8 | 3.5 |
| | CosFace | 20.1 | 17.7 | 1.5 | 0.9 |
| General- izable Models | IBN-Net | 21.4 | 18.9 | 1.5 | 0.8 |
| | TransMatcher | 56.6 | 54.5 | 2.7 | 1.4 |
| | QAConv-GS | 55.1 | 53.0 | 1.2 | 0.6 |
| **Ours** | VAE Embed. | 68.2 | 67.1 | 0.2 | 0.1 |
| | Linear Trans. VAE | 80.7 | 79.2 | 0.4 | 0.2 |

features of images that contain more content or information. However, this method cannot be directly used in our scenario because the query and reference here do not have a simple relationship in terms of information amount. Nevertheless, it offers a promising research direction from the perspective of information. Specifically, on one hand, the noise-adding and denoising processes result in a loss of information, while on the other hand, the guided text introduces new information into the final output.

## F  LIMITATIONS AND FUTURE WORKS

**Limitations.** Although the paradigm analyzed in the main paper (Fig. 13 (a)) is the simplest approach for text-guided image-to-image translation and serves as the default mode in the `AutoPipelineForImage2Image` of `diffusers`, we also observe the existence of an alternative paradigm, as shown in Fig. 13 (b). While this paradigm lies beyond our theoretical guarantees, we can still analyze it experimentally, as demonstrated in Table 9. Interestingly, we find that **(1)** our method generalizes well to InstructP2P, which still uses a *VAE encoder* to embed the original images; and **(2)** all methods, including ours, fail on **IP-Adapter**, which uses *CLIP for encoding*. We also try the linear transformed CLIP embedding, but it still fails to generalize (36.6% mAP and 27.8% Acc). Based on these experiments, we conclude with a hypothesis about the upper limit of our method:

**Hypothesis 1.** *Following **Theorem 1**, consider a different well-trained diffusion model $\mathcal{F}_3$ and its text-guided image-to-image functionability achieved with **VAE-encoded original images**. The matrix $\mathbf{W}$ can be generalized such that for any original image $o$ and its translation $g_3$, we have:*

$$\mathcal{E}_1(g_3) \cdot \mathbf{W} = \mathcal{E}_1(o) \cdot \mathbf{W}. \tag{22}$$

**Future Works.** Future works may focus on **(1)** providing a theoretical proof for Hypothesis 1, and **(2)** developing new generalization methods for text-guided image-to-image based on CLIP encodings.

