# OpenReview forum: "Generalizable Origin Identification for Text-Guided Image-to-Image Diffusion Models"
_ICLR.cc/2025/Conference — ICLR 2025 Conference Withdrawn Submission_

### Official Review · Reviewer_qEy4 · 2024-10-27

**Soundness:** 3
**Presentation:** 3
**Contribution:** 2
**Rating:** 5
**Confidence:** 3

**Summary:**

This paper addresses the task of origin identification for text-guided image-to-image diffusion models (ID2), which transform images based on textual descriptions. Such models, while powerful, pose security concerns like spreading misinformation, copyright infringement, and content evasion. The authors propose a solution to identify the original image from a modified query by introducing the first dataset, OriPID, and a theoretically guaranteed method that highlights generalizability. They prove the existence of a linear transformation that minimizes the distance between VAE-encoded embeddings of generated images and their originals, demonstrating that this transformation generalizes across different diffusion models. Experimental results show their method significantly outperforms traditional similarity-based approaches, effectively addressing the challenge of identifying origins across diverse diffusion models.

**Strengths:**

### Strengths

#### Originality
The paper introduces a novel and timely task—origin identification for text-guided image-to-image diffusion models (ID2). This problem formulation is unique and well-justified, addressing pressing security concerns associated with the misuse of these powerful models. The authors' approach to proving theoretical guarantees for linear transformations applied to VAE embeddings is both creative and original. This innovative combination of diffusion models and theoretical linear algebra provides a robust framework for tackling the ID2 task.

#### Quality
The quality of the research is high, with well-designed experiments and rigorous theoretical analysis. The authors systematically demonstrate the limitations of existing methods and provide comprehensive empirical evidence for the effectiveness and generalizability of their proposed approach. Their method significantly outperforms traditional similarity-based techniques, even those designed with domain generalization in mind, which highlights the robustness and applicability of their solution.

#### Clarity
The paper is clearly written and well-structured. The problem formulation, methodology, and experimental results are presented in a logical and coherent manner. The inclusion of detailed theoretical proofs and thorough explanations of experimental setups further enhances understanding. Figures and tables are effectively used to illustrate key points and results. For instance, the step-by-step explanation of how to learn the theoretical-expected matrix $W$ using gradient descent is particularly clear and helpful.

#### Significance
This research addresses significant real-world issues, such as misinformation, copyright infringement, and content tracing evasion, by targeting the misuse of advanced generative models. By providing a generalized and theoretically sound solution for identifying the origins of modified images, this work has substantial implications for enhancing security and trustworthiness in digital media. The introduction of the OriPID dataset and the associated method sets a new benchmark for future research in this area, offering a valuable resource for the community.

**Weaknesses:**

- While the paper demonstrates impressive generalizability across several contemporary diffusion models, it is unclear how the proposed method would perform with future models that may employ significantly different architectures or embedding mechanisms. The paper could be strengthened by including a discussion on potential limitations of the method with respect to evolving model architectures and providing further analysis or experiments with models that deviate more substantially from those currently tested.

**Questions:**

### Questions and Suggestions

#### 1. Future Diffusion Models
**Question:** How do you anticipate the performance and generalizability of your method evolving as new and potentially very different diffusion models are developed? For instance, models that do not use VAE for feature compression, or those that use Next Token Prediction-based methods for generation, such as MAR: https://arxiv.org/abs/2406.11838.

#### 2. Scalability with Larger Datasets
**Question:** Have you tested your method on significantly larger datasets? How do you ensure that the linear matrix \( W \) can still guarantee performance when the data size becomes very large?

#### 3. Robustness to Adversarial Attacks
**Question:** How does your method perform against sophisticated adversarial attacks designed to fool the VAE embeddings?

In summary, the paper indeed proposes an effective method for retrieving original images after AIGC editing. However, I believe this approach may have considerable limitations.

---

### Official Review · Reviewer_yazG · 2024-11-01

**Soundness:** 3
**Presentation:** 3
**Contribution:** 3
**Rating:** 5
**Confidence:** 4

**Summary:**

This work raises the growing need for identification (ID) preservation in the text-to-image diffusion models for image translation tasks. The authors argue current paradigm exposes several concerns, including the use of misinformation, copyright infringement, and evasion of content tracing. To address these issues, the authors propose a novel task, $ID^2$, aiming to trace a generated image back to its original source query image.

For this task, the authors contribute in two folds: (1) they introduce a new dataset named OriPID, which is derived from DISC21, plus additional GPT-4o generated text prompts for both training and testing; (2) the authors also introduce a method based on a simple linear transformation that minimizes discrepancies between embeddings of original and modified images, enabling it to work with different diffusion models. The experiments shows that the method outperforms current similarity-based approaches, including a variety of supervised/self-supervised/multimodal models, achieving a 31.6% higher mean average precision (mAP) in identifying original images, highlighting its potential for robust origin identification in real-world applications.

**Strengths:**

- The topic discussed in the paper is interesting. The challenge of ID preservation requires ways to trace modified images back to their original versions, which is crucial for accountability, especially in detecting and managing misinformation and copyright issues.

- The paper is overall organized. Additionally, the proposed method is technically sound and the theoretical results look correct.

- The experimental results well validate the approach. Notably, the method demonstrates a siginificant improvement, justifying more reliable origin identification.

- The constructed data is scalable with public foundation models.

**Weaknesses:**

- As one core contribution of the paper, the construction of the dataset is not specific and sufficiently informative to the readers. For example, the 100k origins are randomly selected from DISC21. This could result in reproducibility issues. The authors may need to provide accurate meta information of the dataset or consider to release the dataset in the future.

- Although the authors claim the dataset is scalable, the authors may need to provide specific instructions on how to scale the dataset with LLMs. Do the authors conduct additional operations in selecting the generated 20 prompts or post-processing? Moreover, the authors may need to provide more detailed hyper-parameters when prompting the LLMs and text-to-image diffusion models, e.g., how many NFEs, the CFG scales.

- It is unclear how Lemma 1 support the proof of Theorem 1, as Lemma 1 holds true if Theorem 1 holds. Similarly, it is not clear to me how Lemma 2-4 support the proof of Theorem 2. The authors claim the proof is based on Observation 1. However, it seems we will have such observation almost for sure if the statement of Theorem 2 holds true, while it is not necessary to have Theorem 2 if we have Observation 1. The authors may need to provide more justification on how to justificate Theorem 1 and 2.

- The authors only prove the existence of the matrix $W$. It would be more rigorous if the authors could further discuss additional conditions for a complete analysis, such as the uniqueness.

- The proof of Lemma 2 - 4 seems to be very similar to the results in Homogeneous Systems, SVD and rank-related knowledge in linear algebra. Please correct me if I am wrong. Otherwise I am not sure this could be counted as the contribution of this paper. Can the authors clarify the differences?

**Questions:**

Please see the weaknesses.

**Details Of Ethics Concerns:**

This paper contains a new dataset constructed by the authors and some text are generated using LLMs. It could involves legal and safety issues.

---

### Official Review · Reviewer_zEoK · 2024-11-03

**Soundness:** 2
**Presentation:** 2
**Contribution:** 2
**Rating:** 5
**Confidence:** 4

**Summary:**

This work proposes $ID^2$, a new task (dataset and method) for retrieving the original source image for any given transformed image using an Image-to_image diffusion model.  The authors introduce a large-scale dataset, OriPID, that includes 100,000 source images and transformations of those images across various diffusion models, providing a resource for training and evaluating $ID^2$. the authors emphasize the existence of a linear transformation of the VAE embeddings of an origin image and its translation such that their distance is close enough. They also emphasize generalizability that shows the type of VAE being used does not affect their method's success.

**Strengths:**

This work is well motivated and continues in the direction of copy detection to retrieve the source of a given I2I edited query image given a pool of images. The theorems and lemmas shows the existence of a linear translation of the VAE embeddings of an origin image and its translation such that their distance is close enough and using their created dataset, learns a linear transformation matrix $W$ that enables the alignment of image embeddings across different diffusion models. The experiments show improvement compared to some of the recent copy detection works and also acknowledges the shortcomings against methods SUCH AS IP-Adapter, which uses CLIP-based encodings instead of VAEs.

**Weaknesses:**

- The tone of the paper and the writing could be improved. At times, the paper has redundancy and repetitiveness (explaining the role of $W$, discussion on generalization, etc.)
- From the paper and the proofs, is it fair to assume that the model would work if the translation goes through two different diffusion models? Origin $\rightarrow$ SDXL $\rightarrow$ Kandinsky-3 ? Would be interesting to mention such cases.
- The observation of different VAE embeddings not affecting the performance of the method is interesting. It does make sense that the high dimensionality of VAE latent and the general tendency to capture similar high-level semantic features is leading to such behaviour. However, what happens if the boundaries are pushed on how these VAEs are trained? if there is a large domain gap such as one VAE having been trined on medical images while the other has been trained on portraits? Albeit this might not be practical, but the effects it would have on the assumptions and the empirical results of the paper would be interesting to see.

**Questions:**

My last two weakness points pose my two main questions regarding the paper. I am interested to hear the authors' thoughts on whether these are valid concerns and points that could improve the overall strength of the paper's claims.

---

### Official Review · Reviewer_aJqz · 2024-11-09

**Soundness:** 2
**Presentation:** 3
**Contribution:** 2
**Rating:** 5
**Confidence:** 3

**Summary:**

This paper proposes a new task, namely origin IDentification for text-guided Image-to-image Diffusion models  (ID2), which identifies the original image given a diffusion model translated query. Due to the existence of visual descrepancy issue, solving this task is non-trivial. To this end, this paper contributes the first dataset and also comes with a theoretically guaranteed linear transformation method. Experiments show that the proposed mthod exhibit good generalization performance over images generated by seven different text-to-image diffusion models.

**Strengths:**

1. The task of identifying the original image of a diffusion model translated one is new.
2. The proposed method is conceptually simple and has a theoretically guaranteed optimal solution.
3. The proposed method shows strong performances compared with other baselines.

**Weaknesses:**

1. It looks like in the task of ID2, to find the original image,the proposed method needs to compute the similarity of the query image and each reference image. Thus, the scalability of this approach is limited, because (1) the user has to ensure the original image is already in the reference set, (2) the feature of each reference image needs to be extracted.
2. This task itself is not clearly defined for certain samples. Often times even in the eyes of an human observer (like myself), the generated image bears little-to-no visual similarity to the original image. Take the images in figure 2 of this paper as an example. The dog image generated by Stable Diffusion 3 is very different original dog image (only face vs whole body, facing readers vs facing left, black and white vs all black, etc). In other words, it is hard to confidently say the generated image originated from the original image because of the visual disparity. And I fail to see related efforts by the authors to overcome this issue in dataset curation or task definition.

**Questions:**

1. How well does the proposed method generalize to other T2I diffusion models? I see that Open DALL-e is included in this paper. However, the authors should also try closed-source commercial models like Dall-E 3 or MidJourney. Even a small scale study would enhance this work, considering the cost of these commercial models. Also, strong open-source models like PixArt-Alpha/Delta should also be considered.
2. The proposed method looks a bit too simple to me (although having a theretical optimal solution is nice). Training on diffusion model extracted features is a intuitive solution. I wonder how other baseline models perform if they are also trained on diffusion model extracted feature, rather than raw pixels.

---

### Note · Authors · 2024-11-13

I have read and agree with the venue's withdrawal policy on behalf of myself and my co-authors.